# Unravelling Insights into the Evolution and Management of SARS-CoV-2

Aganze Gloire-Aimé Mushebenge [1,2,3,*], Samuel Chima Ugbaja [2,4], Nonkululeko Avril Mbatha [4,5], Rene B. Khan [2] and Hezekiel M. Kumalo [2,*]

1   Discipline of Pharmaceutical Sciences, University of KwaZulu-Natal, Westville, Durban 4000, South Africa
2   Drug Research and Innovation Unit, Discipline of Medical Biochemistry, School of Laboratory Medicine and Medical Science, University of KwaZulu-Natal, Durban 4000, South Africa
3   Faculty of Pharmaceutical Sciences, University of Lubumbashi, Lubumbashi 1825, Democratic Republic of the Congo
4   Africa Health Research Institute, University of KwaZulu-Natal, Durban 4000, South Africa
5   Department of Human Physiology, School of Laboratory Medicine and Medical Science, University of KwaZulu-Natal, Durban 4000, South Africa
*   Correspondence: aganzedar@gmail.com (A.G.-A.M.); kumaloh@ukzn.ac.za (H.M.K.)

**Abstract:** Worldwide, the COVID-19 pandemic, caused by the brand-new coronavirus SARS-CoV-2, has claimed a sizable number of lives. The virus' rapid spread and impact on every facet of human existence necessitate a continuous and dynamic examination of its biology and management. Despite this urgency, COVID-19 does not currently have any particular antiviral treatments. As a result, scientists are concentrating on repurposing existing antiviral medications or creating brand-new ones. This comprehensive review seeks to provide an in-depth exploration of our current understanding of SARS-CoV-2, starting with an analysis of its prevalence, pathology, and evolutionary trends. In doing so, the review aims to clarify the complex network of factors that have contributed to the varying case fatality rates observed in different geographic areas. In this work, we explore the complex world of SARS-CoV-2 mutations and their implications for vaccine efficacy and therapeutic interventions. The dynamic viral landscape of the pandemic poses a significant challenge, leading scientists to investigate the genetic foundations of the virus and the mechanisms underlying these genetic alterations. Numerous hypotheses have been proposed as the pandemic has developed, covering various subjects like the selection pressures driving mutation, the possibility of vaccine escape, and the consequences for clinical therapy. Furthermore, this review will shed light on current clinical trials investigating novel medicines and vaccine development, including the promising field of drug repurposing, providing a window into the changing field of treatment approaches. This study provides a comprehensive understanding of the virus by compiling the huge and evolving body of knowledge on SARS-CoV-2, highlighting its complexities and implications for public health, and igniting additional investigation into the control of this unprecedented global health disaster.

**Keywords:** SARS-CoV-2; COVID-19; mutations; vaccines; therapeutics; drug repurposing; management strategies

## 1. Introduction

The pandemic caused by the outbreak of severe acute respiratory syndrome coronavirus 2 (SARS-CoV-2) in late 2019 caused a considerable amount of illness and high mortality rates globally [1,2]. Coronaviruses, a group of RNA viruses, cause various diseases in humans and animals and exhibit a characteristic crown-like shape under an electron microscope [3]. These viruses are zoonotic and are transmitted from animals to humans [4]. Notable among them are SARS-CoV, Middle East respiratory syndrome (MERS-CoV), and SARS-CoV-2 [3,5]. SARS-CoV initiated a global outbreak in 2003, with over 8000 cases and significant mortality [6]. MERS-CoV, identified in 2012, primarily affects the Middle

East with a higher mortality rate [7]. SARS-CoV-2, which causes COVID-19, emerged in Wuhan, China, in late 2019, leading to a widespread global pandemic [8]. All three viruses share zoonotic origins, posing significant challenges to public health. SARS-CoV and MERS-CoV have been managed through measures such as quarantine, contact tracing, and travel restrictions, while no specific treatment or vaccine exists for MERS-CoV, making it an ongoing concern, particularly in the Middle East [9]. SARS-CoV-2, the most recent coronavirus, demonstrates a zoonotic origin, possibly from bats and an intermediate host like a pangolin, before infecting humans [10].

SARS-CoV-2's fast spread and mutation have posed a serious challenge to global public health and medical communities [11]. Since its appearance, extensive efforts have been made to understand how SARS-CoV-2 is transmitted, its clinical presentation, and treatment strategies [12]. Understanding the evolution and epidemiology of the virus has laid the groundwork for the creation of solutions to stop its spread and lessen the pandemic's negative effects on public health and the global economy [13].

The continuing evolution of the SARS-CoV-2 virus hampered significantly the creation of efficient prevention and management plans [14]. To provide efficient treatments and vaccines, scientists and researchers from all over the world have made enormous efforts to understand the mechanics of viral reproduction and mutation [15]. A number of vaccines with high success rates against SARS-CoV-2 have been developed as a result of these efforts, and several clinical trials are ongoing to investigate additional therapeutic possibilities [16].

Due to the disease's variable clinical symptoms and uncertain course, SARS-CoV-2 care still faces significant hurdles [17]. Fever, coughing, and shortness of breath are the most typical signs of COVID-19. However, the virus can also cause severe respiratory distress, pneumonia, and multi-organ failure [18,19]. Management of COVID-19 requires a multidisciplinary approach, including supportive care, oxygen therapy, and antiviral treatments [20].

With severe morbidity and mortality, and extensive interruptions to everyday life and economic activities, SARS-CoV-2 has had an unprecedented worldwide impact [21]. The pandemic has made it clear how crucial it is for nations to work together and have a coordinated public health response for facing new infectious illnesses [22]. A thorough knowledge of the virus's evolution, transmission, and clinical symptoms is necessary for the creation of efficient SARS-CoV-2 preventive and management measures.

This work intends to offer a summary of the current knowledge about SARS-CoV-2, including its evolution (see Figure 1), transmission, clinical manifestation, and management, in this in-depth review. This study reviews the most recent findings on the processes of viral replication, mutation, and transmission and the status of available therapies and vaccine research. In addition, we will explore the global impact of the pandemic and the public health response to mitigate the spread of the virus. This paper contributes to the existing literature on SARS-CoV-2 and is organised as follows: clinical description of SARS-CoV-2, SARS-CoV-2 prevalence and pathology, discussion on postulated hypothesis on COVID-19 mutations, COVID-19 therapies, vaccines, and other ongoing clinical trials therapies, drug repurposing for COVID-19, non-pharmacological interventions, and finally a conclusion and future perspectives for COVID-19 research.

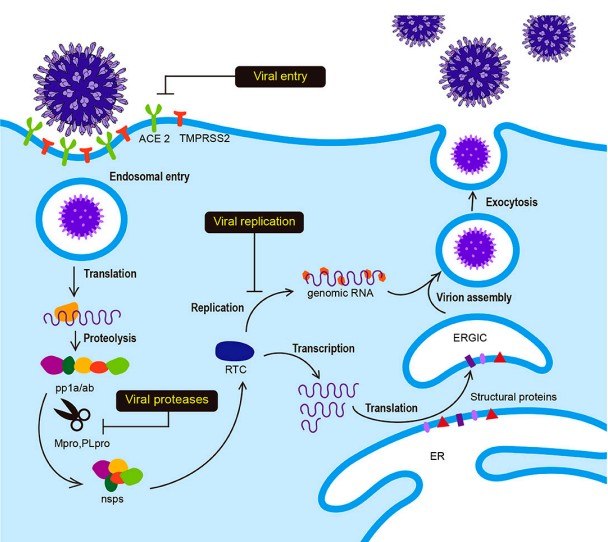

**Figure 1.** Redrawn viral life cycle of SARS-CoV-2 via BioRender as adapted from the source: the S protein's interaction with hACE2 initiates the SARS-CoV-2 viral life cycle and causes infection. Proteolytic cleavage occurring after receptor engagement enables the viral entrance and release of genomic RNA into the cytoplasm. Polyprotein translation is initiated by RNA and is broken into non-structural proteins by enzymes such as PLpro and Mpro. The replicase—transcriptase complex, which is responsible for genome synthesis—is composed of non-structural proteins. Once produced, structural proteins travel to the endoplasmic reticulum where they assemble with RNA encapsulated in N proteins. Once the newly created virions have broken through the cell membrane, they are exocytosis [23].

## 2. Clinical Description of SARS-CoV-2

Despite containment efforts, the COVID-19 pandemic, which started in Wuhan, China, in December 2019, spread rapidly throughout the world [24,25]. The respiratory system is the primary target of COVID-19 symptoms, which frequently include a dry cough, acute respiratory discomfort, pneumonia, and tiredness. A fever is present in virtually all cases [26]. As with its predecessors SARS-CoV and MERS-CoV, COVID-19 is brought on by SARS-CoV-2, which has caused a global health emergency and high rates of morbidity and mortality [27]. As of 2 February 2024, COVID-19 has had more than 702 million confirmed cases and 6.9 million fatalities with a regular daily updated status on the Worldometer coronavirus monitoring instrument [28,29]. The World Health Organization classified COVID-19 as a Public Health Emergency of International Concern because of the pandemic's intensity [30].

In addition to the above typical clinical manifestations of the virus, some patients may also exhibit unusual symptoms such as diarrhoea, headaches, and myalgia [26,31,32]. The condition can be mild or catastrophic in severity, and some patients require mechanical breathing and intensive care [32,33]. Secondary infections such as bacterial pneumonia, which possibly worsen patient outcomes, can affect the clinical course of COVID-19 [34].

Understanding the clinical characteristics of the illness is essential for providing appropriate medical care and creating successful preventative and treatment plans as the COVID-19 pandemic spreads. Healthcare specialists throughout the world are working non-stop to contain the pandemic because of the severity of the illness and its potential for spread, which have made it a global health emergency. Critical elements in the fight against COVID-19 include quick diagnostic tests, sufficient personal protective equipment for healthcare workers, and efficient vaccinations and antiviral treatments. To guide public health policy and stop the spread of the disease, more studies are needed on the clinical manifestation and pathogenesis of SARS-CoV-2.

## 3. SARS-CoV-2 Prevalence and Pathology

Coronaviruses such as SARS-CoV, MERS-CoV, and SARS-CoV-2 were first discovered in bats and spread to humans via intermediate hosts such as civet cats and camels [35]. The first instances of human-to-animal transmission were recorded in Hong Kong in February 2020, but the COVID-19 pandemic has shown that companion animals such as cats and dogs can also be susceptible to the virus [36]. In addition, SARS-CoV-2 has been detected in monkeys, white-tailed deer, and minks, indicating a wider spectrum of possible hosts. Numerous animal species, including ferrets, hamsters, macaques, and baboons, have been shown susceptibility to SARS-CoV or SARS-CoV-2 infection in experimental infection studies [37,38].

### 3.1. SARS-CoV-2 Transmission, Clinical Presentation, and Risk Factors for Severe Disease and Fatality

There have been reports of gastrointestinal involvement, and the discovery of viral RNA in faecal samples raises the possibility of faecal–oral transmission [39]. Isolation and medical monitoring are essential preventive measures because asymptomatic carriers can unintentionally spread the infection to others [39,40]. A median hospital stay of 12 days is required for 20% of confirmed COVID-19 patients, and 25% of patients in hospitals require acute critical care [40]. Adults 55 years and older are the main demographic for severe COVID-19 cases. Mortality risk increases gradually, with a mortality rate of 1.4–4.9% in the 55–74 age group, 4.3–10.5% in the 75–84 age group, and 10.4–27.3% in the 85-plus age group [41].

According to the available data, SARS-CoV-2 is a naturally occurring virus that is mainly spread by inhaling cough droplets [42]. Another important method of transmission occurs when hands that have come into contact with droplet-contaminated surfaces touch the face, eyes, or nose [42,43]. Despite the lack of certainty regarding the seasonality of SARS-CoV-2, mounting evidence points to a potential role for climate factors in the spread of the virus [44].

There are three stages of SARS-CoV-2 clinical pathology: mild, severe, and critical, with critical being the stage that results in mortality [45]. Adults who are infected typically show no symptoms or only mild, temporary symptoms, whereas those who show symptoms are most contagious the day before symptoms appear [46,47].

With a typical incubation time of 4–6 days, COVID-19 clinical signs include respiratory and intestinal problems [48]. It is difficult to detect transmission chains and conduct subsequent tracing because the clinical symptoms are less severe than those associated with SARS and MERS infections [49]. Severe COVID-19 symptoms are more likely to develop in older, immune-compromised people with pre-existing diseases such as cardiovascular disease, hypertension, asthma, and diabetes [37,50].

There have been reports of gastrointestinal involvement, and the discovery of viral RNA in faecal samples raises the possibility of faecal–oral transmission [51,52]. Isolation and medical monitoring are essential preventative measures because asymptomatic carriers can unintentionally spread the infection to others [51]. To stop the virus's cycle of spread, preventive measures were implemented, including limiting population movements, preventing large gatherings, and closing educational institutions [53].

Some individuals who recovered from COVID-19 had severe lung damage by the 10th day after the onset of symptoms [54]. In addition, pneumonia linked to SARS-CoV-2 infection was observed to include a significant portion of the lower respiratory tract [55]. Contrary to SARS-CoV and MERS-CoV, studies indicate that SARS-CoV-2 infection during pregnancy does not result in maternal fatalities, and there is no proof that the virus is transmitted to unborn children either intrauterinally or transplacentally [56].

Furthermore, long COVID, or post-acute sequelae of SARS-CoV-2 infection (PCC), is a prevalent condition that occurs months after COVID-19 infection and is characterised by symptoms such as fatigue, dyspnoea, memory loss, diffuse pain, and orthostasis [57]. PCC has significant medical, psychosocial, and economic impacts, leading to widespread

unemployment and substantial lost wages [57,58]. Pathophysiological mechanisms involve central nervous system inflammation, viral reservoirs, persistent spike protein, cell receptor dysregulation, and autoimmunity [59]. In long-term COVID, symptoms persisted for up to two years post-infection [60]. This symptomatology is heterogeneous, and potential mechanisms include viral persistence, inflammation, immune dysregulation, autoimmune reactions, latent infections, endothelial dysfunction, and alterations in gut microbiota [60,61]. Healthcare systems are grappling with the social significance of post-COVID syndrome, emphasising the need for dynamic patient follow-up and rehabilitation programmes [62]. The U.S. CDC acknowledges symptoms lasting over four weeks as "Post-COVID Condition," but the precise pathogenesis remains multifactorial. It involves organ dysfunction, immune system responses, and the effects of severe illness, causing microvascular injury and abnormalities in organ functioning [63]. Long-term symptoms, affecting approximately 10% of COVID survivors, necessitate a comprehensive rehabilitation approach to improve various body systems' functions [64].

### 3.2. Profile Characteristics and Prognostic Markers of COVID-19/SARS-CoV-2

Aspartate aminotransferase and hypersensitive troponin I levels are high in COVID-19 individuals, and they also exhibit unique blood laboratory profile traits such as lymphopenia, leukopoenia, thrombocytopenia, and RNAaemia [65]. Procalcitonin levels begin at a normal range but gradually increase as the disease progresses, indicating an elevated risk of secondary infections [65,66]. Erythrocyte sedimentation rate (ESR) and C-reactive protein (CRP) levels have increased in COVID-19, whereas platelet count and procalcitonin levels are normally within the range. Aspartate aminotransferase (AST), alanine transaminase (ALT), lactate dehydrogenase (LDH), creatine phosphokinase (CPK), creatinine, and prothrombin time, which can be employed as diagnostic markers, are associated with higher levels in severe instances (see Figure 2) [65–67].

Prognostic indicators for COVID-19 include lymphopenia, neutrophilia, increased levels of LDH, C-reactive protein, D-dimer, total bilirubin, hepatic transaminases, ferritin, and troponins [68]. The numbers of T cells, B cells, and natural killer (NK) cells are also lowered in severe instances, as are the numbers of helpers, regulatory, and memory T cells. Severe instances are marked by higher levels of proinflammatory cytokines and chemokines such as IFN-, IL-1, IP-10, MCP-1, TNF-, G-CSF, IL-8, IL-10, and MIP-1A [67,68].

Unlike SARS and MERS, in which only Th1 cytokines are elevated, COVID-19 shows an elevation of both Th1 and Th2 cytokines [69]. The release of pro-IL-1b and subsequent synthesis of mature IL-1b, which regulates fever, pulmonary inflammation, and fibrosis, are triggered by the interaction of SARS-CoV-2 with Toll-like receptors (TLR) [70]. Therefore, IL-37 and IL-38 could be considered as appropriate therapeutic agents and may be highly beneficial in COVID-19 patients to reduce pulmonary inflammation by suppressing IL-1b and other proinflammatory IL-family members [71]. Different HLA types and varying epitope binding affinities influenced various immunopathological effects induced by the novel human coronavirus COVID-19 in humans [72].

Genomic and microsatellite analyses identified 1191 protein targets related to ACE2, revealing 305 host factors associated with COVID-19/asthma comorbidity [73]. Enrichment analyses highlighted the significance of metabolic processes, Th1 and Th2 cell differentiation, and PPAR signalling pathways in this comorbidity. Key host factors, including HRAS, IFNG, CAT, CDH1, FASN, ACLY, CCL5, VCAM1, SCD, and HMGCR, were identified, with some highly expressed in lung tissues [74–76]. In the exploration of genetic variations in the ACE2 and TMPRSS2 receptor genes in COVID-19 patients, there were associations between specific genotypes and expressions of these genes with SARS-CoV-2 positivity, including clinical outcomes [77]. The identification of haplotypes and variants/mutations contributes to the understanding of host genetic factors influencing COVID-19 susceptibility, offering insights for future vaccine development and therapeutic approaches [78,79].

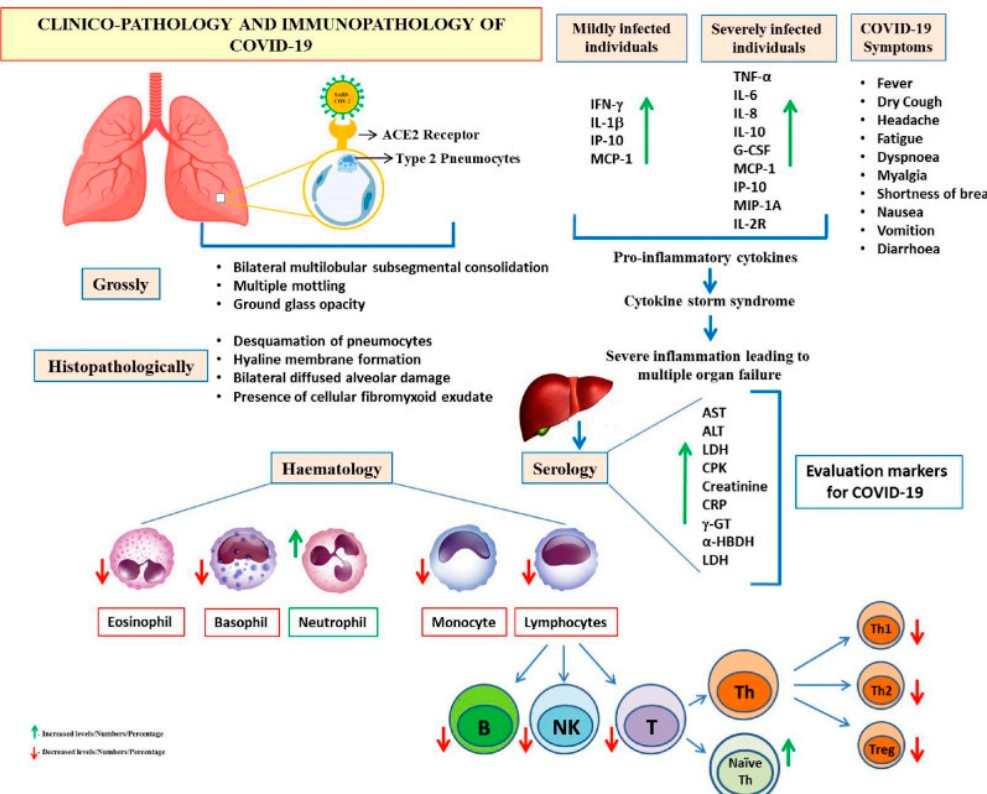

**Figure 2.** An overview of the clinical pathology, pathogenesis, and immunopathology of COVID-19 as adapted from the source: the virus impacts the immune system by modifying the production of biomolecules in immune cells during disease progression. The clinical signs of COVID-19 can range from digestive to pulmonary issues, and the disease has a 5.99% case fatality rate. Asymptomatic carriers can shed the virus for up to 21 days. Fever, coughing, exhaustion, and respiratory distress are examples of clinical symptoms. The cytokine storm, humoral and cellular processes, and lymphopenia are all components of the immune response. The virus primarily causes cytopathic effects on epithelial cells in the gastrointestinal and respiratory systems. COVID-19 patients have different lymphocyte subsets, cytokine upregulation, and immunopathologically altered leukocyte numbers [12].

## 4. Discussion on the Postulated Hypothesis on COVID-19 Mutations

Periods of relative stability followed by the emergence of variants marked the evolution of COVID-19. In an unprecedented effort to stop the COVID-19 epidemic, the scientific community from around the world has teamed up [80]. Similar to other RNA viruses, COVID-19 exhibits a high mutation rate that can be caused by copying errors during viral replication, recombination, or contact with agents that can neutralise the virus, such as host antibodies [81,82]. The transport and load of the virus to ACE2 target cells may be improved by molecular changes that lessen the damage to the viral capsid, which is crucial for safeguarding the viral genome and replication [82]. This would increase infectivity without necessarily changing the virus' inherent pathogenicity or virulence. Mutations that increase the affinity of spike S-protein to receptors on ACE2 cells could also increase viral load and infectivity [83].

In the variants of concern, Omicron (B.1.1.529) played a pivotal role in influencing infectivity with numerous mutations in the receptor-binding domain (RBD). The Omicron lineage, characterised by numerous sublineages, including BA.1, BA.2, BA.4, and BA.5, rapidly became globally dominant, with variations in the receptor-binding domain (RBD) of the spike protein. These variants raise concerns regarding vaccine effectiveness and potential reinfections [84]. Despite these mutations, the Omicron RBD binds to the human ACE2 (hACE2) receptor with a similar affinity as the prototype RBD, possibly due to compensatory mutations [85]. Key residues identified, including Q493, Q498, N501,

F486, K417, and F456, are crucial for tight binding to hACE2 [86]. In the Omicron variant, mutated residues R493, S496, and R498 formed new interactions with ACE2, compensating for the reduced ACE2 binding affinity, leading to similar biochemical ACE2 binding affinities for Delta and Omicron [87,88]. These mutations contribute to increased infectivity and transmissibility, accompanied by a notable ability to evade neutralising antibodies, highlighting the complex interplay between spike protein mutations and viral characteristics [87–91]. This molecular understanding could inform the development of therapeutic and prophylactic agents targeting these variants.

Furthermore, the D614G mutation in the SARS-CoV-2 spike protein was discovered early in the epidemic and quickly became the predominant form everywhere [54,55]. The D614G mutation in the SARS-CoV-2 spike protein enhances viral transmission and infectivity [92,93]. This mutation is associated with increased binding to the human cell-surface receptor ACE2, leading to heightened replication in primary human airway epithelial cultures, human ACE2 knock-in mice, and enhanced replication and transmissibility in hamster and ferret models [92,94]. The variant containing S(D614G) exhibited a very competitive advantage during the transmission bottleneck, explaining its global predominance [94–96]. Population genetic analysis indicates a selective advantage for the 614G variant, reflected in increased frequency, higher viral load, and a younger age of patients. This supports its role in improved transmission without indicating higher mortality or clinical severity [97,98].

The mutation improves the virus's capacity to infect cells and may increase its transmissibility [99]. Other changes to the spike protein could make the virus more resistant to antibodies produced by the host immune system or vaccines, thus decreasing their effectiveness [100]. Therefore, it is essential to track SARS-CoV-2 mutations to spot any potentially dangerous variants and create workable defences (see Figure 3).

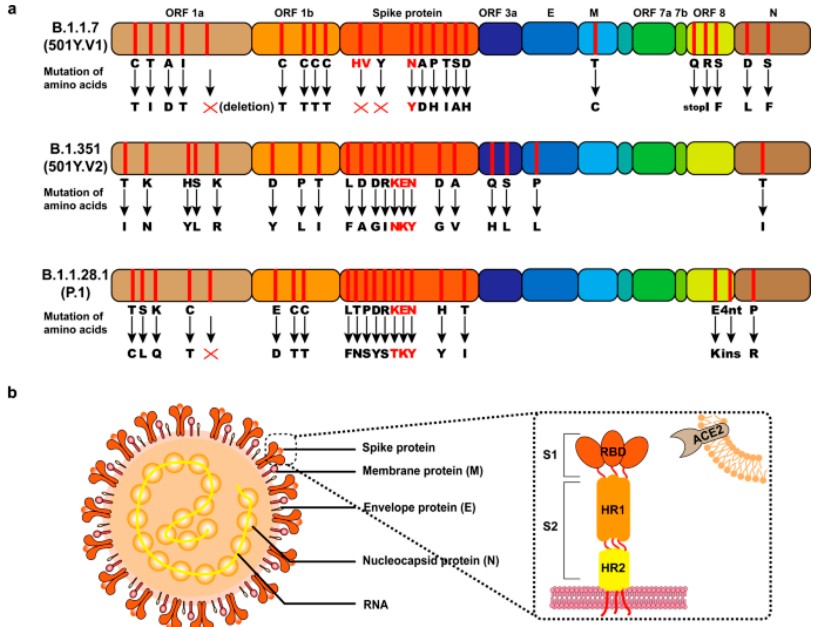

**Figure 3.** Information of fast-spreading SARS-CoV-2 variants and major SARS-CoV-2 structures as adapted from the source: the illustration shows the precise amino acid mutations that are present in SARS-CoV-2 variations that are spreading quickly. (**a**) Highlights the important mutations in variants like B.1.1.7, B.1.351, and B.1.1.28.1 by highlighting them in red. (**b**) Spike protein, membrane protein, envelope protein, nucleocapsid protein, and RNA—the main structural elements of SARS-CoV-2—are shown. Angiotensin-converting enzyme 2 (ACE2) is emphasised as the main cellular entry point for SARS-CoV-2, highlighting its function as a transmembrane protein [101].

Studying and comprehending the underlying principles of viral evolution is essential given the rapid evolution of SARS-CoV-2 and the high mutation rates observed [98–100,102]. This information will be useful in creating efficient vaccinations and therapies that can be quickly modified to address newly discovered virus strains. The progress achieved thus far emphasises the need for ongoing multidisciplinary research and collaboration to contain the ongoing COVID-19 pandemic [101]. The scientific community has made tremendous progress in understanding the biology of SARS-CoV-2.

## 5. Advancements in SARS-CoV-2/COVID-19 Management

### 5.1. COVID-19 Therapies, Vaccines, and Other Ongoing Clinical Trial Therapies

Effective treatments and vaccines are urgently needed to combat the COVID-19 pandemic (see Table 1). It is crucial to create decoy ligands with structure-based designs that can obstruct crucial infectivity-mediating activities such as docking, binding, entrance, and replication [103]. While natural supplements such as vitamin D and zinc are being investigated for their potential prophylactic benefits, supercomputers have been used to screen the library of already licenced medications for potential therapeutics against COVID-19 [104]. Daily vitamin D supplements were found to protect against acute respiratory infections in a recent meta-analysis [105], whereas increased intracellular zinc concentrations have been demonstrated to hinder the reproduction of several RNA viruses [106]. However, excessive intake of zinc can lead to undesirable sequelae, and it is not currently recommended to give elemental zinc supplementation above the recommended dietary allowance for the prevention of COVID-19 [106,107].

Previous studies have shown reduced neutralisation of Omicron by postvaccination serum, indicating the need for ongoing surveillance of genetic and antigenic changes [96]. Monoclonal antibodies, a promising therapeutic for COVID-19, face challenges in effectiveness due to evolving spike mutations [108]. The Omicron variant, with 37 amino acid substitutions in the spike protein, exhibits enhanced affinity for the ACE2 receptor, causing marked reductions in neutralising activity against Omicron compared with the ancestral virus [109]. The emergence of variants, including Alpha, Beta, Gamma, and Delta, has further complicated the pandemic landscape, necessitating additional research avenues to understand their impact on transmission, reinfection risk, and vaccine protection [110,111]. The challenges of long-lasting symptoms in a young patient emphasise the need for therapeutic approaches such as heparin-induced extracorporeal LDL/fibrinogen precipitation (H.E.L.P.) apheresis [112].

Moreover, the COVA trial revealed that 20-hydroxyecdysone (BIO101) demonstrated statistically significant efficacy, showing a 43.8% reduction in the risk of death or respiratory failure ($p = 0.0426$), with nine patients needing treatment. The results confirm BIO101's good safety profile and support its clinical relevance in modulating the renin– angiotensin system through MASR activation for treating COVID-19 [113,114]. These findings indicate the need for further investigation of BIO101 as a potential treatment for hospitalised patients with severe pneumonia due to COVID-19 [113].

**Table 1.** Summary of current potential anti-SARS-CoV-2 agents.

**Approved SARS-CoV-2 Vaccines**

| Vaccine Name | Manufacturer | Type | Dosage/Post-dosage | Efficacy | Target | Ref. |
|---|---|---|---|---|---|---|
| BNT162b2 | Pfizer, BioNTech | mRNA | Two doses, 4–8 weeks apart. A booster (4–6 months after). | 95% | Spike protein | [115,116] |
| mRNA-1273 | Moderna | mRNA | Two doses, 4–8 weeks apart. Dose 3 at least 4 weeks after Dose 2. A single booster dose (0.25 mL) may be administered at least 5 months after completing a primary series. | 94.1% | Spike protein | [115,117] |
| Ad26.COV2.S | Johnson and Johnson | Viral vector | One dose (preferred). Administration of the second dose to increase level of protection against symptomatic infection. W.H.O recommends interval of 2 to 6 months. Booster dose after 90 days. | 72% (in the U.S.) | Spike protein | [115,118] |
| ChAdOx1-S [recombinant] | AstraZeneca, University of Oxford | Viral vector | Two doses, with an interval of 8 to 12 weeks. A booster dose may be considered 4–6 months after completion of the primary vaccination series. | 70.4% (average) | Spike protein | [115,119] |

**SARS-CoV-2 therapeutics**

| Drug name | Manufacturer | Type | Target | Antiviral Agent | Status | Ref. |
|---|---|---|---|---|---|---|
| Remdesivir | Gilead Sciences | Antiviral | RNA polymerase | Nucleotide analogue | FDA-approved for emergency use in hospitalised patients | [120] |
| Baricitinib | Eli Lilly and Company | Anti-inflammatory | AP-1 | Janus kinase inhibitor | FDA-approved for emergency use along with remdesivir | [121,122] |
| Tocilizumab | Roche | Anti-inflammatory | IL-6 | The monoclonal antibody (mAb) | FDA-approved for emergency use in hospitalised patients | [123,124] |
| Sotrovimab | GlaxoSmithKline and Vir Biotechnology | The monoclonal antibody (mAb) | Spike protein | The monoclonal antibody (mAb) | FDA-approved for emergency use in high-risk individuals | [125,126] |
| Molnupiravir | Merck & Co. | Antiviral | RNA polymerase | Nucleotide analogue | Currently under review for emergency use authorisation | [127,128] |

**Table 1.** *Cont.*

| Ongoing Clinical Trials for SARS-CoV-2 | | | | | | |
|---|---|---|---|---|---|---|
| **Study name** | **Sponsor** | **Type** | **Phase** | **Target** | **Antiviral Agent/Status** | **Ref.** |
| ACTIV-6 | NIH | Therapeutics | 3 | Various | Various/Ongoing | [129,130] |
| COMET-ICE | NIAID, Lilly | Therapeutics | 3 | Various | Various/Ongoing | [126,131] |
| REGN-COV2 | Regeneron | Therapeutics | 3 | Spike protein | The monoclonal antibody (mAb)/Ongoing | [132,133] |
| COV-BOOST | University of Oxford | Vaccine | 2 | Spike protein | N/A/Ongoing | [134] |
| COV-FLU | Novavax | Vaccine | 3 | Influenza virus | N/A/Ongoing | [135] |

*5.2. Drug Repurposing for COVID-19*

Drug repurposing has become a vital tactic in the pursuit of successful COVID-19 treatment [93,136,137]. It includes looking at old antiviral medications and substances that are either approved or being researched for use against other viral infections [136]. The World Health Organization conducted the multinational clinical trial SOLIDARITY to investigate the efficacy of repurposed medications in treating COVID-19 [138]. Drugs such as lopinavir–ritonavir, chloroquine, remdesivir, and favipiravir are among those being investigated. The possible applications of these medications for treating COVID-19 will be discussed in this article [139].

The FDA has approved the medication combination Lopinavir–Ritonavir for the treatment of HIV-1. Ritonavir boosts the efficacy of lopinavir by delaying the rate at which it is metabolised in the liver, whereas lopinavir is a protease inhibitor that prevents virus particle formation [140]. The medication has shown some promise in treating COVID-19, but more clinical trial findings are needed to confirm its effectiveness [136,140].

An RNA-dependent RNA polymerase inhibitor called favipiravir has shown promise in combating influenza and other viral diseases [141]. Initial clinical trials conducted in Shenzhen and Wuhan have demonstrated its efficacy against SARS-CoV-2 [142]. Favipiravir-treated patients had a stronger therapeutic response, especially in terms of faster viral clearance and a higher rate of improvement in chest imaging [143]. The National Medical Products Administration of China has approved favipiravir as the first anti-COVID-19 medication in the nation considering these encouraging findings [142,143].

Moderately priced medications such as chloroquine and hydroxychloroquine, which are used to treat autoimmune disorders and malaria, are thought to reduce endosomal pH, preventing SARS-CoV-2 replication [43,144,145]. Although chloroquine (CQ) and hydroxychloroquine (HCQ) have shown in vitro inhibition of coronaviruses such as SARS-CoV and SARS-CoV-2, clinical study results have been inconsistent, indicating an incomplete understanding of their antiviral mechanisms [146,147]. Research indicates that HCQ dose dependently inhibits the ACE2 enzyme, but structural changes from the ACE2–S protein interaction may affect HCQ binding [148–151]. Despite FDA emergency use authorisation for COVID-19 treatment, it was revoked on 15 June 2020 because clinical studies demonstrated ineffectiveness in reducing death rates or shortening recovery time for hospitalised patients. These findings align with those of other studies questioning the ability of HCQ to inhibit or eliminate SARS-CoV-2 [152,153].

The nucleotide analogue prodrug Remdesivir provides broad-spectrum antiviral action against various RNA viruses [102]. In contrast to protease inhibitors, which focus on the late steps of virus reproduction, RNA-dependent RNA polymerase inhibits the early stage of viral replication [103]. It has been used as an experimental medicine for the treatment of Ebola, MERS-CoV, and SARS-CoV-2 and has been demonstrated to limit the replication of SARS-CoV-2 in animal models [104]. Clinical improvement was observed in 68% of patients treated with Remdesivir in a sample of critically ill patients hospitalised for severe COVID-19 [105,106]. A study revealed that remdesivir-treated patients, mostly men in their fifth decade with comorbidities, had a lower mortality risk, reduced incidence of ARDS, and lower ventilator dependency compared with the control group [154].The drug demonstrated effectiveness in mitigating kidney involvement caused by SARS-CoV-2, with no significant adverse effects reported [154,155].

Furthermore, a study explored drug repurposing as a cost-effective strategy for SARS-CoV-2 treatment by docking 16 antiviral approved drugs against the protease protein (6M03) responsible for viral replication [156]. Delavirdine, fosamprenavir, imiquimod, stavudine, and zanamivir exhibited excellent results, indicating their potentially strong drug profiles [156]. The isoquinoline derivative SLL-0197800, known for its anticoronaviral activities, was tested for its inhibitory effects on RNA-dependent RNA polymerase (RdRp) and 3′-to-5′ exoribonuclease (ExoN) proteins [157]. In vitro assays demonstrated potent inhibitory activities with low EC50 values (0.16 and 0.27 µM, respectively). In silico results supported these findings, indicating the potential of SLL-0197800 against various coronavi-

ral strains, including future versions, such as SARS emphasising, emphasising its specific anticoronaviral qualities. This study underscores the potential of drug repurposing and highlights SLL-0197800 as a promising candidate for broader coronaviral infections [157].

In addition, therapeutic options have been sought in natural products (terpenoids, alkaloids, saponins, and phenolics) with promising in vitro and in silico results for use in COVID-19 disease [158–160]. Among these, the most studied are resveratrol, quercetin, hesperidin, curcumin, myricetin, and betulinic acid, which were proposed as SARS-CoV-2 inhibitors [159]. Regarding natural products, resveratrol, curcumin, and quercetin have demonstrated in vitro antiviral activity against SARS-CoV-2, and in vivo, a nebulised formulation has been demonstrated to alleviate the respiratory symptoms of COVID-19 [158,159,161].

### 5.2.1. Vaccine Development and Challenges

Scientists worldwide are engaged in a race to develop effective vaccines against SARS-CoV-2, using traditional and next-generation platforms. Current COVID-19 vaccines induce immune responses against the viral spike protein or its subunits, thereby preventing cellular entry [162]. The emergence of SARS-CoV-2 variants poses challenges, prompting the design of a peptide-based vaccine with epitopes targeting pathogenic proteins [45]. Computational methods, reverse vaccinology, and immunoinformatics guide the creation of a multi-epitope-based peptide vaccine by considering mutations in spike glycoprotein. Immune profiling, codon optimisation, and in silico simulations contribute to vaccine development [163].

Inactivated and attenuated vaccines, protein subunit and virus-like particle vaccines, viral vector-based vaccinations, and more recent DNA- and RNA-based vaccines are a few of the techniques for vaccine creation that are being studied [164]. All approaches are being developed simultaneously to create an effective vaccine, and each has advantages and disadvantages (Figure 4). Because it is present in all coronaviruses encountered and is exposed to the immune system of a person, the spike protein is thought to be the most promising vaccine among the structured proteins of the virus because it allows the body to mount an immune response against it and retain it for future defence [165].

Notably, all FDA-approved vaccines effectively generate neutralising antibodies. The Pfizer–BioNTech and Moderna mRNA vaccines, among others, have received global approval, reflecting ongoing efforts to combat the evolving virus [163,166,167]. Several vaccines have been approved in different parts of the world, such as CoronaVac, BBIBP-CorV, CoviVac, Covaxin, Oxford–AstraZeneca vaccine (ChAdOx1 nCoV-19), Sputnik V, the Johnson & Johnson vaccine, Convidicea, RBD-Dimer, and EpiVacCorona [168,169].

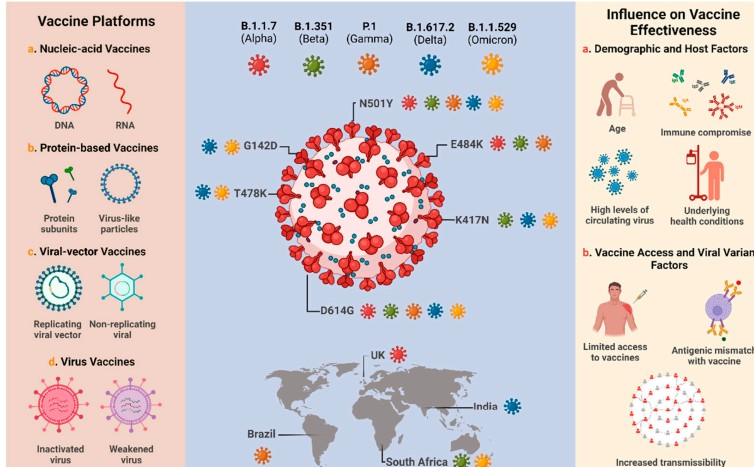

**Figure 4.** An overview of the main platforms for technology employed in the creation of the COVID-19 vaccine, the SARS-CoV-2 variants of concern, the corresponding changes in their spike proteins, and the variables that could affect the efficacy of the vaccines already on the market. Created with BioRender.com as adapted from source [170].

The mRNA-1273 vaccine, a revolutionary RNA-based vaccine created by Moderna Therapeutics, was the first to start clinical testing [171]. To inject nanoparticles into the body, it exploits a portion of the spike protein genetic code [171,172]. The first phase I clinical trial of this vaccine, which had shown promise in animal research, began on March 16, 2020, in partnership with the National Institutes of Health (NIH), and involved 45 healthy people between the ages of 18 and 55 [172].

There are numerous additional mRNA-based vaccines in various stages of development [173]. To address viral infection, further cutting-edge vaccine strategies and treatment interventions are being explored. For instance, Codagenix developed a live-attenuated vaccine employing a reverse technique by changing the viral sequences by substituting its optimised codons with non-optimised ones to weaken the virus [174], and Inovio Pharmaceuticals' INO-4800 is a DNA-based vaccination using the spike gene [175]. The Milken Institute COVID-19 Treatment and Vaccine Tracker provides information on several vaccinations currently being developed, along with their present progress [176]. Significant international vaccine funding organisations supported numerous innovative initiatives to select the most promising vaccines for future mass production.

Subsequently, a study examined the impact of different treatments on SARS-CoV-2 antibody levels in patients with autoimmune rheumatic disease undergoing Pfizer/BioNTech mRNA vaccination [177]. Original adalimumab treatment resulted in stable antibody levels, whereas biosimilar adalimumab showed a significant decrease [177]. This study underscores the dominant influence of treatment type on antibody changes over time, with negligible contributions from other variables [178]. In the context of mRNA vaccines, a booster administered 3–4 weeks after the initial vaccination substantially increased protective antibody levels [179]. Ipsilateral boost of the SARS-CoV-2 mRNA vaccine induced more germinal centre B cells specific to the receptor-binding domain (RBD), generating increased bone marrow plasma cells compared with the contralateral boost. An ipsilateral boost also rapidly produced high-affinity RBD-specific antibodies with enhanced cross-reactivity to the Omicron variant [179]. Evaluating T-cell responses in patients with late-stage lung cancer receiving immune-modulating agents, including anti-PD-1/PD-L1, showed distinct CD8+ T-cell responses with only marginal CD4+ T-cell responses to the Omicron variant. This emphasises the need for heightened protective measures for cancer patients because of qualitative deviations in T-cell responses [180].

### 5.2.2. Experimental Therapeutic Interventions
#### Convalescent Plasma (CP) Therapy

Since the start of the pandemic, there has been active research into experimental therapeutic interventions for COVID-19. Convalescent plasma (CP) therapy, a conventional method, has been used successfully for more than a century to treat infectious diseases such as SARS, MERS, and H1N1 [181]. To provide this treatment, neutralising antibody-rich plasma from a recovered patient is extracted and given to the infected patient [182]. Significant improvement has been observed in preliminary tests on patients with severe COVID-19, and further clinical trials are still being conducted. Researchers are also profiling specific antibodies from recovered patients to create functional antibodies as COVID-19 treatment, in addition to CP therapy [182,183].

Subsequently, more than 500 distinct antibodies were found in the serum of a recovering COVID-19 patient, and companies such as AbCellera and Eli Lilly are collaborating to create medicines based only on human IgG1 monoclonal antibodies (mAbs) [184]. mAbs targeting the SARS-CoV-2 spike protein, isolated from memory B cells of an individual infected with SARS-CoV in 2003, exhibited potent neutralisation of SARS-CoV-2 and SARS-CoV [185]. The antibody S309, which recognises a conserved epitope containing glycan, demonstrates strong neutralisation without competing with receptor attachment. Antibody cocktails including S309 enhance SARS-CoV-2 neutralisation, potentially limiting the emergence of escape mutants [185,186]. These findings highlight the therapeutic potential of these antibodies, especially S309, for prophylaxis or post-exposure therapy to mitigate

severe COVID-19. In addition, a study identified 11 potent neutralising antibodies against SARS-CoV-2, with antibody 414–1 showing the best IC50 of 1.75 nM, providing promising therapeutic candidates for COVID-19 treatment [185–187].

In addition, cutting-edge methods for treating COVID-19, such as aerosolised siRNAs and nanoviricides, are being developed [188]. A technique created by Alnylam Pharmaceuticals that delivers aerosolised siRNAs directly to the lungs is undergoing in vitro and in vivo testing [189]. On the other hand, "virucidal nanomicelles" are being chemically attached to the S protein to form nanoviricides [190]. Because complement factor 5a is the primary contributor to tissue damage in patients, InflaRx and Beijing Defengrei Biotechnology are developing human IgG1 mAbs against it [189,190]. These antibodies have already received Chinese government approval for clinical trials. These cutting-edge treatments may be able to effectively cure COVID-19 and help to contain the current pandemic [190].

Soluble Human Angiotensin-Converting Enzyme 2

Because of its capacity to prevent SARS-CoV-2 replication, soluble human angiotensin-converting enzyme 2 (ACE2) has become recognised as a potential COVID-19 treatment option [191]. The virus enters human cells through the cellular receptor ACE2; hence, inhibiting the binding between the Spike protein and ACE2 may be a useful therapeutic approach [192]. Recent in vitro investigations have demonstrated the therapeutic potential of human recombinant soluble ACE2 (hrsACE2), which can considerably lower viral loads in Vero cells and block viral infection in constructed human blood arteries and kidney organoids [193,194]. These results indicate that hrsACE2 has the potential to safeguard patients from lung injury and SARS-CoV-2 infection by preventing viral entry into target cells (see Figure 5) [195].

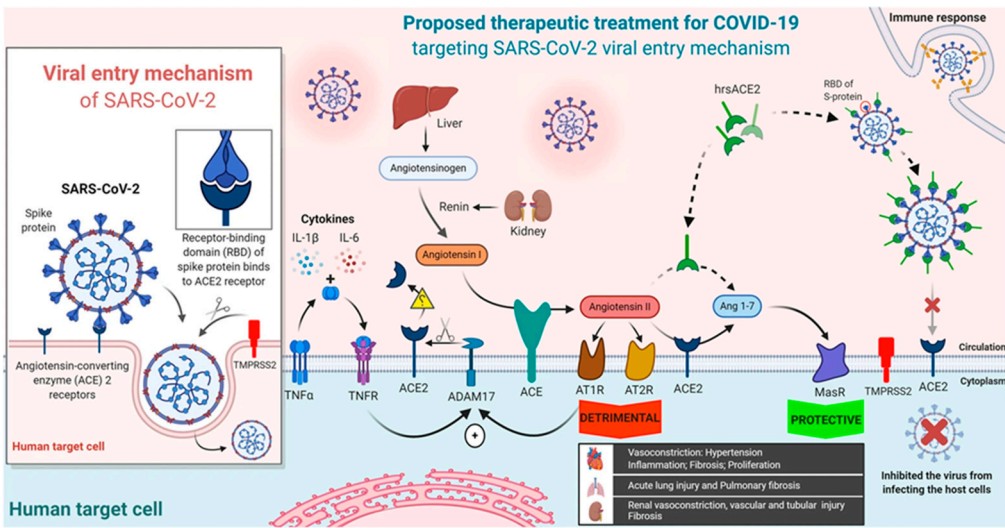

**Figure 5.** Schematic of the renin–angiotensin system, and the proposed treatment for COVID-19 targeting the SARS-CoV-2 viral-entry mechanism as adapted from source: The renin–angiotensin system and a suggested treatment strategy for COVID-19 that targets the SARS-CoV-2 viral-entry mechanism are depicted in this diagram. The receptor-binding domain (RBD) of the spike protein interacts with ACE2 on the left, enabling host cell entrance and infection. The physiological function of ACE2 in the renin–angiotensin system and its organ-protective effects are discussed in the middle. ACE2 hydrolyses Angiotensin II to produce angiotensins 1–7, imparting a protective effect. For good health, ACE/Ang II/AT1R and ACE2/Ang 1-6/MasR must be in equilibrium. According to recent research, increasing the amount of human recombinant soluble ACE2 (hrsACE2) at tissue sites may compete with endogenous ACE2, preventing SARS-CoV-2 from infecting host cells and lowering angiotensin II levels while maintaining the production of anti-SARS-CoV-2 antibodies [196].

In the fight against COVID-19, the use of soluble ACE2 as a therapeutic intervention shows promise [197]. hrsACE2 can prevent viral replication and lower viral loads by preventing communication between the virus and its host receptor [198]. Moreover, hrsACE2 has shown efficacy against SARS-CoV-2 in human blood vessels and kidney organoids, indicating its potential to protect patients from the virus's severe lung damage [108]. Therefore, COVID-19-patient therapy may benefit from the use of soluble ACE2. However, more investigations are required to establish its security and effectiveness in clinical trials [199].

## 6. Non-Pharmacological Interventions

Implementing non-pharmacological interventions (NPIs) was, therefore, the most efficient public health response to the outbreak [38,199,200]. NPIs included early case detection and isolation, in-depth contact tracing of suspected secondary cases, travel prohibitions, tight contact reductions, physical segregation, increased cleanliness, and routine hand washing [201]. Closing non-essential public areas, services, and facilities was one of these strategies. Another is for educational institutions to switch to digital learning modalities and for enterprises to implement self-isolation/work from home programmes [38]. According to modelling projections, integrated NPIs are anticipated to have the biggest and fastest impact on reducing the reproductive number and slowing the rate of viral transmission if they are adopted early in the outbreak [199,202,203]. The creation of efficient treatment interventions and vaccines is made possible using the knowledge gained from these NPIs, which are interim measures while the effort to better understand viral genomes continues [38,201].

To reduce transmission, such measures include early detection of infected persons, seclusion, and tracking of their close connexions. Travel restrictions, social withdrawal, and enhanced hygiene helped to stop the virus's spread [204]. NPIs can reduce viral reproduction and delay viral transmission if implemented early, which would reduce the impact of the outbreak. However, while NPIs are effective in the interim, efforts must continue to develop effective therapeutic interventions and vaccines to address the ongoing crisis [205,206].

Despite the challenges posed by the COVID-19 pandemic, maintaining a regular physical activity practise was proven to be beneficial for reducing the likelihood of Musculoskeletal (MSK) pain in infected individuals [187]. These findings are crucial for healthcare professionals managing long-term COVID patients experiencing MSK pain [207,208]. In addition, a study that supports the effectiveness of non-pharmacological strategies and a proposed control system grounded in control system theory aims to guide public decision policies in managing COVID-19 spread and preventing healthcare system overload by anticipating peak infection rates and implementing timely interventions [207–209].

## 7. Conclusions and Future Perspectives for COVID-19 Research

Reducing infections, lowering the strain on healthcare systems, and lessening the pandemic's social and economic effects are the three main goals of efforts to limit the COVID-19 pandemic. Non-pharmacological therapies will continue to serve as the main line of protection while we wait for viable vaccinations. Therefore, projections and planning for anticipated healthcare capacity can be informed by accurate and current data on the daily number of new cases and the case characteristics [210]. The bacille Calmette—Guerin childhood vaccine as well as national immunisation programmes may have an impact on the pandemic intensity. COVID-19 will undoubtedly have a large worldwide impact that could take a long time to reverse [211]. To combat upcoming pandemics, healthcare systems must consider including efficient regulatory mechanisms. The approach to the present pandemic has already been influenced by lessons learned from the earlier SARS-CoV outbreaks in Hong Kong, Singapore, and Taiwan [212,213]. With regard to the pathogenicity, transmissibility, and therapeutic response of the viral isolates, genomic characterisation will also affect regional and global populations.

Additionally, artificial intelligence (AI) plays a crucial role in various aspects of addressing the challenges posed by COVID-19, including modelling and simulation, employing AI robotics for medical quarantine, and predicting the spread of the virus [214] AI systems, encompassing machine learning, deep learning, convolutional neural networks, and cognitive computing, offer significant utility in virus detection, comprehensive screening, continuous monitoring, and alleviating the workload for healthcare providers [199,214–216]. Furthermore, they proved instrumental in anticipating potential interactions with novel treatments. Considering these advancements, we recommend that computer scientists focus their efforts on refining and advancing these AI-driven methods, as they hold promise for effective responses to future outbreaks [38,214,215].

The international community must cooperate to make the greatest technology resources available to combat the current pandemic and prepare for potential future epidemics [217]. For the development of efficient vaccinations and the discovery of new drugs, it is essential to understand the genetic makeup of viral strains [172,218]. Data-driven initiatives should guide planning and predictions for expected healthcare capacity [219]. To combat upcoming pandemics, effective regulatory measures and national immunisation strategies should be considered [38,219]. To anticipate and track infections before an outbreak occurs, AI should be tested. The social, cultural, and economic infrastructures will be significantly impacted by the COVID-19 pandemic over the long term; thus, it is crucial to draw lessons from this experience and improve readiness for future breakouts.

**Author Contributions:** Conceptualization, A.G.-A.M., S.C.U. and H.M.K.; writing—original draft preparation, A.G.-A.M. and S.C.U.; writing—review and editing, A.G.-A.M., S.C.U. and N.A.M.; supervision, H.M.K. and R.B.K. All authors have read and agreed to the published version of the manuscript.

**Funding:** This research received no external funding.

**Institutional Review Board Statement:** Not applicable.

**Informed Consent Statement:** Not applicable.

**Data Availability Statement:** Not applicable.

**Acknowledgments:** The authors would like to thank the College of Health Sciences, University of KwaZulu-Natal for the support.

**Conflicts of Interest:** The authors declare no conflicts of interest.

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
