# Peer review of "Unravelling Insights into the Evolution and Management of SARS-CoV-2"

_biomedinformatics, doi:10.3390/biomedinformatics4010022_

Round 1
Reviewer 1 Report
Comments and Suggestions for Authors
This review aims to provide a comprehensive exploration of our current understanding of SARS-CoV-2. The contents of this review are potentially interesting. However, this review should be major revised to address the following comments:
Major comments:
1) Lines 26-29, the authors mentioned that “the review aims to clarify the complex network of factors that have contributed to the varying case fatality rates observed in different geographic areas. In this effort, we explore the complex world of SARS-CoV-2 mutations and their implications in vaccines efficacy and therapeutic interventions.” However, this review does not have enough information of how the mutations impact vaccines efficacy and therapeutic interventions. Therefore, I suggest adding more details in section 4 about the impacts of mutations of these medical options including the most recent variants of SARS-CoV-2, Omicron. Please try to use the following references:
https://doi.org/10.1038/s41579-021-00573-0; https://doi.org/10.1016/j.cell.2022.06.005; https://doi.org/10.1038/s41586-021-04387-1; https://doi.org/10.1016/j.chom.2022.05.001; https://doi.org/10.1016/j.jiph.2021.12.014; https://doi.org/10.1016/j.xcrm.2021.100255; https://doi.org/10.1016/j.compbiomed.2023.107576; https://doi.org/10.1038/s41586-021-04386-2; https://doi.org/10.1038/s41576-021-00408-x.
Similarly after line 202, give some examples of the mutations on the Spike protein that have the ability to increase the infectivity such as N501Y, etc. Try use the following references:
https://doi.org/10.1016/j.cell.2022.01.001; https://doi.org/10.1038/s41467-021-26401-w; https://doi.org/10.1021/acs.jcim.1c00560; https://doi.org/10.1021/acs.jcim.1c01451;
DOI:10.1126/science.abn7760;
https://doi.org/10.1021/acs.jpclett.2c00423; https://doi.org/10.1016/j.celrep.2022.110729
Additionally after line 207, explain how the D614G mutation improves infectivity and transmission.
2) All figure’s captions (Figure 1 to 5) require more details about the process or the pathway to be easy to read.
Additionally, it should explain what is (a) and (b) in Figure 3.
3) The review contains a lot of repetitive sentences or ideas.
For examples:
a) Sentences in lines 83-85 are similar to sentences in lines 93-96, sentence in line 124, and sentence in line 152. Here, I suggest deleting the sentences in lines 83-85, in line 124, and in line 152.
b) Sentences in lines 133-135 with the ones in lines 149-151. I recommend deleting one of them.
c) Sentences in lines 125-128 are exactly similar to the ones in lines 153-156. Therefore, the one in lines 153-156 must be removed.
d) Paragraphs, lines 237-256, should be removed. Paragraph, lines 295-300, should also be removed.
4) Since the structure and computational studies play an essential role in our fight against COVID-19, the authors should include a section on these subjects to add more value to this review.
5) Updates are needed for some drugs, especially the ones that are no longer functional. For example, the information about chloroquine and hydroxychloroquine in lines 278–285 needs to be updated because the Solidarity Trial was terminated in 2020 due to evidence that they did not reduce the COVID-19 mortality of hospitalized patients. Therefore, the authors should double-check the new data on these two drugs, as well as others, and use the most recent references.
Similarly, the sentence in line 293 “Remdesivir has not yet received approval to treat COVID-19, and more studies are needed to determine its effectiveness” should be updated.
The sentence in lines 322-323 should also be updated or deleted.
The sentences in line 382-384 should be modified. This is also true for the sentences in line 397 and lines 401-404.
6) In subsection 6.2.1 after line 349 add one paragraph about mAbs.
Minor comments:
1) Line 3, the words “: A Comprehensive Review” should be removed.
2) Lines 46-48, the sentence “The coronavirus family, which also contains the viruses causing Middle East respiratory syndrome (MERS) and severe acute respiratory syndrome (SARS), includes SARS-CoV-2.” Should be modified as “The coronavirus family includes the SARS-CoV-2 virus as well as previous viruses that caused Middle East respiratory syndrome (MERS) and severe acute respiratory syndrome (SARS)”.
3) Line 62, change “has proven difficult” to “still faces significant hurdles”.
4) Line 63, delete “the illness brought on by SARS-CoV-2.”.
5) Lines 86-87, change "COVID-19 is brought on by the severe acute respiratory syndrome coronavirus 2 (SARS-CoV-2), which” to “SARS-CoV-2”.
6) Line 128, is there a full stop, period, before “A median”? Check it.
7) Line 160, the word “Most” should change to “Some”.
8) The sentences in lines 183-184 and lines 186-189 should be cited.
9) Line 190, change “2019-nCoV” to “COVID-19”. Additionally, in lines 192, 194, 220, and 220, change “Covid-19” to “COVID-19”.
10) Sections 5 and 6 should be combined and modified as needed.
11) The authors should double check the details in Table 1, especially the number of doses, is there any post dose, the phase, etc.
12) Line 364, change “this connection” to “the binding between Spike protein and ACE2”.
13) Some abbreviations should be defined such as BCG in line 414 (bacille Calmette-Guerin), AI in line 422 (Artificial intelligence),
14) Add full affiliation address in line 443.
1
Comments on the Quality of English Language
Extensive editing of English language required. See my comments.
Author Response
We wish to express our appreciation to the journal and the reviewer for the in-depth constructive comments, suggestions, and corrections, which have greatly improved the quality of our Manuscript. We carefully considered the latest set of comments raised and duly modified the new version of the Manuscript. A handful of grammatical corrections and typos, and a few places were improved for the Manuscript to read better. The Manuscript went through language editing as well.
Reviewer 1:
Major comments:
Comment 1: Lines 26-29, the authors mentioned that “the review aims to clarify the complex network of factors that have contributed to the varying case fatality rates observed in different geographic areas. In this effort, we explore the complex world of SARS-CoV-2 mutations and their implications in vaccines efficacy and therapeutic interventions.” However, this review does not have enough information of how the mutations impact vaccines efficacy and therapeutic interventions. Therefore, I suggest adding more details in section 4 about the impacts of mutations of these medical options including the most recent variants of SARS-CoV-2, Omicron. Please try to use the following references:
https://doi.org/10.1038/s41579-021-00573-0; https://doi.org/10.1016/j.cell.2022.06.005; https://doi.org/10.1038/s41586-021-04387-1; https://doi.org/10.1016/j.chom.2022.05.001; https://doi.org/10.1016/j.jiph.2021.12.014; https://doi.org/10.1016/j.xcrm.2021.100255; https://doi.org/10.1016/j.compbiomed.2023.107576; https://doi.org/10.1038/s41586-021-04386-2; https://doi.org/10.1038/s41576-021-00408-x.
Similarly, after line 202, give some examples of the mutations on the Spike protein that have the ability to increase the infectivity such as N501Y, etc. Try use the following references:
https://doi.org/10.1016/j.cell.2022.01.001; https://doi.org/10.1038/s41467-021-26401-w; https://doi.org/10.1021/acs.jcim.1c00560; https://doi.org/10.1021/acs.jcim.1c01451;
DOI:10.1126/science.abn7760; https://doi.org/10.1021/acs.jpclett.2c00423; https://doi.org/10.1016/j.celrep.2022.110729
Additionally, after line 207, explain how the D614G mutation improves infectivity and transmission.
Response to comment 1: Thank you for the constructive comment on mutations in section 4. The Section improvement has been addressed and summarised as follows: In the variants of concern, Omicron (B.1.1.529) played a pivotal role in influencing infectivity with numerous mutations in the receptor-binding domain (RBD). The Omicron lineage, characterised by numerous sublineages, including BA.1, BA.2, BA.4, and BA.5, rapidly became globally dominant, with variations in the receptor-binding domain (RBD) of the spike protein. These variants raise concerns regarding vaccine effectiveness and potential reinfections[84]. Despite these mutations, the Omicron RBD binds to the human ACE2 (hACE2) receptor with a similar affinity as the prototype RBD, possibly due to compensatory mutations [85]. Key residues identified, including Q493, Q498, N501, F486, K417, and F456, are crucial for tight binding to hACE2 [86]. In the Omicron variant, mutated residues R493, S496, and R498 formed new interactions with ACE2, compensating for the reduced ACE2 binding affinity, leading to similar biochemical ACE2 binding affinities for Delta and Omicron [87,88]. These mutations contribute to increased infectivity and transmissibility, accompanied by a notable ability to evade neutralising antibodies, highlighting the complex interplay between spike protein mutations and viral characteristics [87-91]. This molecular understanding could inform the development of therapeutic and prophylactic agents targeting these variants. See line 262 to line 277 in the updated version of the manuscript.
Line 207, explain how the D614G mutation improves infectivity and transmission: The D614G mutation in the SARS-CoV-2 spike protein enhances viral transmission and infectivity. This mutation is associated with increased binding to the human cell-surface receptor ACE2, leading to heightened replication in primary human airway epithelial cultures, human ACE2 knock-in mice, and enhanced replication and transmissibility in hamster and ferret models. The variant containing S(D614G) exhibits a real competitive advantage during the transmission bottleneck, explaining its global predominance. Population genetic analysis suggests a selective advantage for the 614G variant, reflected in increased frequency, higher viral load, and a younger age of patients, supporting its role in improved transmission without indicating higher mortality or clinical severity. See line 279 to line 289 in the updated version of the manuscript.
Comment 2: All figure’s captions (Figure 1 to 5) require more details about the process or the pathway to be easy to read. Additionally, it should explain what is (a) and (b) in Figure 3.
Response to comment 2: Thank you for the constructive comment. All figures have been explained for them to be self-explanatory as per the reviewer’s comment. See updated version of the manuscript with highlights in green below each figure.
Comment 3) The review contains a lot of repetitive sentences or ideas.
For examples:
- a)Sentences in lines 83-85 are similar to sentences in lines 93-96, sentence in line 124, and sentence in line 152. Here, I suggest deleting the sentences in lines 83-85, in line 124, and in line 152.
- b)Sentences in lines 133-135 with the ones in lines 149-151. I recommend deleting one of them.
- c)Sentences in lines 125-128 are exactly similar to the ones in lines 153-156. Therefore, the one in lines 153-156 must be removed.
- d)Paragraphs, lines 237-256, should be removed. Paragraph, lines 295-300, should also be removed.
Response to comment 3: Thank you for the constructive comment. The repetitive sentences and ideas have been removed for the manuscript to read better.
Comment 4) Since the structure and computational studies play an essential role in our fight against COVID-19, the authors should include a section on these subjects to add more value to this review.
Response to comment 4: Thank you for the constructive comment. Authors have written a few publications on this project that emphasize the contribution of computational studies in finding prospective drugs for SARS-CoV-2. However, Section 6 of this manuscript has given an overview of the contribution of Biocomputational studies in Drug Discovery to eradicate SARS-CoV-2.
Comment 5) Updates are needed for some drugs, especially the ones that are no longer functional. For example, the information about chloroquine and hydroxychloroquine in lines 278–285 needs to be updated because the Solidarity Trial was terminated in 2020 due to evidence that they did not reduce the COVID-19 mortality of hospitalized patients. Therefore, the authors should double-check the new data on these two drugs, as well as others, and use the most recent references.
Similarly, the sentence in line 293 “Remdesivir has not yet received approval to treat COVID-19, and more studies are needed to determine its effectiveness” should be updated.
The sentence in lines 322-323 should also be updated or deleted.
The sentences in line 382-384 should be modified. This is also true for the sentences in line 397 and lines 401-404.
Response to comment 5: Thank you for the constructive comment. The information about chloroquine and hydroxychloroquine in lines 278–285 has been updated where applicable with the latest references. It reads as follows: Although chloroquine (CQ) and hydroxychloroquine (HCQ) have shown in vitro inhibition of coronaviruses such as SARS-CoV and SARS-CoV-2, clinical study results have been inconsistent, indicating an incomplete understanding of their antiviral mechanisms [146,147]. Research indicates that HCQ dose-dependently inhibits the ACE2 enzyme, but structural changes from the ACE2–S protein interaction may affect HCQ binding [148-151]. Despite FDA emergency use authorisation for COVID-19 treatment, it was revoked on June 15, 2020 because clinical studies demonstrated ineffectiveness in reducing death rates or shortening recovery time for hospitalised patients. These findings align with those of other studies questioning the ability of HCQ to inhibit or eliminate SARS-CoV-2. See line 370 to line 379 in the updated version of the manuscript.
- Zapata-Cardona MI, Flórez-Álvarez L, Zapata-Builes W, et al. Atorvastatin effectively inhibits ancestral and two emerging variants of SARS-CoV-2 in vitro. Front Microbiol. 2022;13(March). doi:10.3389/fmicb.2022.721103
- Marín-Palma D, Tabares-Guevara JH, Zapata-Cardona MI, et al. Curcumin inhibits in vitro sars-cov-2 infection in Vero E6 cells through multiple antiviral mechanisms. Molecules. 2021;26(22):1–17. doi:10.3390/molecules26226900
- Yao X, Ye F, Zhang M, et al. In vitro antiviral activity and projection of optimized dosing design of hydroxychloroquine for the treatment of severe acute respiratory syndrome main point: hydroxychloroquine was found to be more potent than chloroquine at inhibiting SARS-CoV-2 in vit. Clin Infect Dis. 2020;2:1–25.
- Yepes-Perez AF, Herrera-Calderón O, Oliveros CA, et al. The Hydroalcoholic Extract of Uncaria tomentosa (Cat’s Claw) Inhibits the Infection of Severe Acute Respiratory Syndrome Coronavirus 2 (SARS-CoV-2) in Vitro. Evidence-Based Complement Altern Med. 2021;2021:1–11. doi:10.1155/2021/6679761
- U.S Food and Drug Administration (FDA). Memorandum explaining basis for revocation of emergency use authorization for chloroquine phosphate and hydroxychloroquine sulfate. FDA site; 2020. Available from: https://www.fda.gov/news-events/press-announcements/coronavirus-covid-19-update-fda-revokes-emergency-use-authorization-chloroquine-and. Accessed December 18, 2023.
- U.S Food and Drug Administration (FDA). FDA cautions against use of hydroxychloroquine or chloroquine for COVID-19 outside of the hospital setting or a clinical trial due to risk of heart rhythm problems; 2020. Available from: https://www.fda.gov/drugs/drug-safety-and-availability/fda-cautions-against-use-hydroxychloroquine-or-chloroquine-covid-19-outside-hospital-setting-or. Accessed December 18, 2023.
The line 293 on Remdesivir has been updated and reads as follows: Clinical improvement was observed in 68% of patients treated with Remdesivir in a sample of critically ill patients hospitalised for severe COVID-19 [105,106]. A study revealed that remdesivir-treated patients, mostly men in their fifth decade with comorbidities, had a lower mortality risk, reduced incidence of ARDS, and lower ventilator dependency compared with the control group [154].The drug demonstrated effectiveness in mitigating kidney involvement caused by SARS-CoV-2, with no significant adverse effects reported [154,155]. See line 387 to line 391 in the updated version of the manuscript.
- Figueredo, J., Lopez, L.F., Leguizamon, B.F., Samudio, M., Pederzani, M., Apelt, F.F., Añazco, P., Caballero, R. and Bianco, H., 2024. Clinical evolution and mortality of critically ill patients with SARS-CoV-2 pneumonia treated with remdesivir in an adult intensive care unit of Paraguay. BMC Infectious Diseases, 24(1), p.37.
The sentences in line 382-384 have been modified. This is also true for the sentences in line 397 and lines 401-404: Subsequently, a study examined the impact of different treatments on SARS-CoV-2 antibody levels in patients with autoimmune rheumatic disease undergoing Pfizer/BioNTech mRNA vaccination [177]. Original adalimumab treatment resulted in stable antibody levels, whereas biosimilar adalimumab showed a significant decrease [177]. This study underscores the dominant influence of treatment type on antibody changes over time, with negligible contributions from other variables [178]. In the context of mRNA vaccines, a booster administered 3–4 weeks after the initial vaccination substantially increased protective antibody levels [179]. Ipsilateral boost of the SARS-CoV-2 mRNA vaccine induced more germinal centre B cells specific to the receptor binding domain (RBD), generating increased bone marrow plasma cells compared with the contralateral boost. Ipsilateral boost also rapidly produced high-affinity RBD-specific antibodies with enhanced cross-reactivity to the Omicron variant [179]. Evaluating T-cell responses in patients with late-stage lung cancer receiving immune-modulating agents, including anti-PD-1/PD-L1, showed distinct CD8+ T-cell responses with only marginal CD4+ T-cell responses to the Omicron variant. This emphasises the need for heightened protective measures for cancer patients because of qualitative deviations in T-cell responses [180]. . See line 458 to line 473 in the updated version of the manuscript.
Comment 6) In subsection 6.2.1 after line 349 add one paragraph about mAbs.
Response to comment 6: Thank you for the constructive comment. The paragraph about mAbs has been updated as follows: mAbs targeting the SARS-CoV-2 spike protein, isolated from memory B cells of an individual infected with SARS-CoV in 2003, exhibited potent neutralisation of SARS-CoV-2 and SARS-CoV [185]. The antibody S309, which recognises a conserved epitope containing glycan, demonstrates strong neutralisation without competing with receptor attachment. Antibody cocktails, including S309, enhance SARS-CoV-2 neutralisation, potentially limiting the emergence of escape mutants [185,186]. These findings highlight the therapeutic potential of these antibodies, especially S309, for prophylaxis or post-exposure therapy to mitigate severe COVID-19. In addition, a study identified 11 potent neutralising antibodies against SARS-CoV-2, with antibody 414–1 showing the best IC50 of 1.75 nM, providing promising therapeutic candidates for COVID-19 treatment [185-187]. See line 488 to line 497 in the updated version of the manuscript.
Minor comments:
Comment 1) Line 3, the words “: A Comprehensive Review” should be removed.
Response to comment 1: Thank you for the constructive comment. The above-mentioned words have been removed in the updated version of the manuscript. See line 2 to 3.
Comment 2) Lines 46-48, the sentence “The coronavirus family, which also contains the viruses causing Middle East respiratory syndrome (MERS) and severe acute respiratory syndrome (SARS), includes SARS-CoV-2.” Should be modified as “The coronavirus family includes the SARS-CoV-2 virus as well as previous viruses that caused Middle East respiratory syndrome (MERS) and severe acute respiratory syndrome (SARS)”.
Response to comment 2: Thank you for the constructive comment. The issue has been addressed and reads better now. See line 46 to line 59 in the updated version of the manuscript.
Comment 3) Line 62, change “has proven difficult” to “still faces significant hurdles”.
Response to comment 3: Thank you for the constructive comment. The issue has been addressed and reads better now. See line 74 of the updated version of the manuscript.
Comment 4) Line 63, delete “the illness brought on by SARS-CoV-2.”.
Response to comment 4: Thank you for the constructive comment. The above-mentioned words have been removed in the updated version of the manuscript to read better.
Comment 5) Lines 86-87, change "COVID-19 is brought on by the severe acute respiratory syndrome coronavirus 2 (SARS-CoV-2), which” to “SARS-CoV-2”.
Response to comment 5: Thank you for the constructive comment. The issue has been addressed and reads better now.
Comment 6) Line 128, is there a full stop, period, before “A median”? Check it.
Response to comment 6: Thank you for the constructive comment. See line 159 in the updated version of the manuscript.
Comment 7) Line 160, the word “Most” should change to “Some”.
Response to comment 7: Thank you for the constructive comment. See line 187 in the updated version of the manuscript.
Comment 8) The sentences in lines 183-184 and lines 186-189 should be cited.
Response to comment 8: Thank you for the constructive comment. All cited passages have been referenced accordingly in the updated version of the manuscript. See line 230 and line 235.
Comment 9) Line 190, change “2019-nCoV” to “COVID-19”. Additionally, in lines 192, 194, 220, and 220, change “Covid-19” to “COVID-19”.
Response to comment 9: Thank you for the constructive comment. The changes have been made. See lines 237, 250, 254 in the updated version of the manuscript
Comment 10) Sections 5 and 6 should be combined and modified as needed.
Response to comment 10: Thank you for the constructive comment. The issue has been addressed and reads better now. The sections have been combined into one and modifications made in the updated version of the manuscript. See lines 310-538
Comment 11) The authors should double check the details in Table 1, especially the number of doses, is there any post dose, the phase, etc.
Response to comment 11: Thank you for the constructive comment. The table has been updated and read better now as per the reviewer’s comment. See line 344 to line 345 in updated version of the table.
Comment 12) Line 364, change “this connection” to “the binding between Spike protein and ACE2”.
Response to comment 12: Thank you for the constructive comment. The issue has been addressed and reads better now. See line 512 in updated version of the manuscript.
Comment 13) Some abbreviations should be defined such as BCG in line 414 (bacille Calmette-Guerin), AI in line 422 (Artificial intelligence),
Response to comment 13: Thank you for the constructive comment. The issue has been addressed and reads better now. See lines 584 and 594 in updated version of the manuscript.
Comment 14) Add full affiliation address in line 443.
Response to comment 14: Thank you for the constructive comment. The issue has been addressed and reads better now. See lines 613-614 in updated version of the manuscript.
Reviewer 2 Report
Comments and Suggestions for Authors
Dear Author,
I found the manuscript titled “Unraveling Insights into the Evolution and Management of SARS-CoV-2: A Comprehensive Review” to be interesting; however, it lacks new information and could be significantly improved. Please find my comments below.
Major comments:
- The topic of Long COVID, or Post-COVID, is currently of great interest to readers, yet it is absent from the manuscript. The document should include details on the pathology, symptoms, treatments, and management of Long COVID.
- The manuscript also needs to be updated to reflect the latest findings on Long COVID.
Minor comments:
1. It lacks bioinformatics or computational content to be considered in this journal. Maybe choice of journal could be re-considered.
Kind regards,
Comments on the Quality of English Language
Minor editing of English language required
Author Response
We wish to express our appreciation to the journal and the reviewer for the in-depth constructive comments, suggestions, and corrections, which have greatly improved the quality of our Manuscript. We carefully considered the latest set of comments raised and duly modified the new version of the Manuscript. A handful of grammatical corrections and typos, and a few places were improved for the Manuscript to read better. The Manuscript went through language editing as well.
Reviewer 2:
I found the manuscript titled “Unravelling Insights into the Evolution and Management of SARS-CoV-2: A Comprehensive Review” to be interesting; however, it lacks new information and could be significantly improved. Please find my comments below.
Major comments:
Comment 1. The topic of Long COVID, or Post-COVID, is currently of great interest to readers, yet it is absent from the manuscript. The document should include details on the pathology, symptoms, treatments, and management of Long COVID. The manuscript also needs to be updated to reflect the latest findings on Long COVID.
Response to comment 1: Thank you for the constructive comment. The issue has been addressed and reads better now. The latest findings on Long COVID/Post-COVID: pathology: Furthermore, long COVID, or post-acute sequelae of SARS-CoV-2 infection (PCC), is a prevalent condition that occurs months after COVID-19 infection and is characterised by symptoms such as fatigue, dyspnoea, memory loss, diffuse pain, and orthostasis [57]. PCC has significant medical, psychosocial, and economic impacts, leading to widespread unemployment and substantial lost wages [57,58]. Pathophysiological mechanisms involve central nervous system inflammation, viral reservoirs, persistent spike protein, cell receptor dysregulation, and autoimmunity [59]. In long-term COVID, symptoms persisted for up to two years post-infection [60]. This symptomatology is heterogeneous, and potential mechanisms include viral persistence, inflammation, immune dysregulation, autoimmune reactions, latent infections, endothelial dysfunction, and alterations in gut microbiota [60,61]. Healthcare systems are grappling with the social significance of post-COVID syndrome, emphasising the need for dynamic patient follow-up and rehabilitation programmes [62]. The U.S. CDC acknowledges symptoms lasting over four weeks as "Post-Covid Condition," but the precise pathogenesis remains multifactorial. It involves organ dysfunction, immune system responses, and the effects of severe illness, causing microvascular injury and abnormalities in organ functioning [63]. Long-term symptoms, affecting approximately 10% of COVID survivors, necessitate a comprehensive rehabilitation approach to improve various body systems’ functions See line 193 to line 210 in the updated version of the manuscript.
The latest findings on Long COVID/Post-COVID: treatments, and management:
Previous studies have shown reduced neutralisation of Omicron by postvaccination serum, indicating the need for ongoing surveillance of genetic and antigenic changes [96]. Monoclonal antibodies, a promising therapeutic for COVID-19, face challenges in effectiveness due to evolving spike mutations [108]. The Omicron variant, with 37 amino acid substitutions in the spike protein, exhibits enhanced affinity for the ACE2 receptor, causing marked reductions in neutralising activity against Omicron compared with the ancestral virus [109]. The emergence of variants, including Alpha, Beta, Gamma, and Delta, has further complicated the pandemic landscape, necessitating additional research avenues to understand their impact on transmission, reinfection risk, and vaccine protection [110,111]. The challenges of long-lasting symptoms in a young patient emphasise the need for therapeutic approaches such as heparin-induced extracorporeal LDL/fibrinogen precipitation (H.E.L.P.) apheresis [112]. Moreover, the COVA trial revealed that 20-hydroxyecdysone (BIO101) demonstrated statistically significant efficacy, showing a 43.8% reduction in the risk of death or respiratory failure (p = 0.0426), with a number of patients needed to treat 9. The results confirm BIO101’s good safety profile and support its clinical relevance in modulating the renin– angiotensin system through MASR activation for treating COVID-19 [113,114]. These findings indicate the need for further investigation of BIO101 as a potential treatment for hospitalised patients with severe pneumonia due to COVID-19 [113]. See line 325 to line 343 in the updated version of the manuscript.
Lobo, S.M., Plantefève, G., Nair, G., Cavalcante, A.J., de Moraes, N.F., Nunes, E., Barnum, O., Stadnik, C.M.B., Lima, M.P., Lins, M. and Hajjar, L.A., 2024. Efficacy of oral 20-hydroxyecdysone (BIO101), a MAS receptor activator, in adults with severe COVID-19 (COVA): a randomized, placebo-controlled, phase 2/3 trial. eClinicalMedicine.
Minor comments:
Comment 1. It lacks bioinformatics or computational content to be considered in this journal. Maybe choice of journal could be re-considered.
Response to comment 1: Thank you for the comment. The paper did not detail much on this aspect but has a whole section (6) about the biocomputational chemistry via drug repurposing. See line 344 to line 409 in the updated version of the manuscript. However, this aspect has been further discussed in our previous publications as part of the project.
Reviewer 3 Report
Comments and Suggestions for Authors
I have the following suggestions to modify the manuscript.
1. There are various types of vaccines, such as Herbal and Antibiotic. Can you please add literature references regarding these vaccine types?
2. Many studies are available on Drug Repurposing with Deep Learning. Please include a literature table highlighting this perspective.
3. Elaborate in the paper on the major vaccines for SARS-Covid-19, addressing both mutation and normal proteins.
4. Add the structure of the paper at the end of the introduction.
5. All the images appear to be copied directly. Have you obtained permission to use them in your paper?
6. It would be beneficial to include some drugs that are highly predicted for SARS-COVID-19 in recent studies.
7. Provide a proper introduction to the Covid family, including SARS-CoV, MERS-CoV, SARS-CoV-2, and BAT-CoV, among others.
8. Include analysis studies such as genome analysis and microsatellite analysis in your paper.
Comments on the Quality of English LanguageModerate correction required
Author Response
We wish to express our appreciation to the journal and the reviewer for the in-depth constructive comments, suggestions, and corrections, which have greatly improved the quality of our Manuscript. We carefully considered the latest set of comments raised and duly modified the new version of the Manuscript. A handful of grammatical corrections and typos, and a few places were improved for the Manuscript to read better. The Manuscript went through language editing as well.
Reviewer 3:
Comment 1. There are various types of vaccines, such as Herbal and Antibiotic. Can you please add literature references regarding these vaccine types?
Response to comment 1: Thank you for the constructive comment. The issue has been addressed and reads as follows: A study revealed that remdesivir-treated patients, mostly men in their fifth decade with comorbidities, had a lower mortality risk, reduced incidence of ARDS, and lower ventilator dependency compared with the control group [154].The drug demonstrated effectiveness in mitigating kidney involvement caused by SARS-CoV-2, with no significant adverse effects reported [154,155].
Furthermore, a study explored drug repurposing as a cost-effective strategy for SARS-CoV-2 treatment by docking 16 antiviral approved drugs against the protease protein (6M03) responsible for viral replication [156]. Delavirdine, fosamprenavir, imiquimod, stavudine, and zanamivir exhibited excellent results, indicating their potential as strong profile drugs [156]. The isoquinoline derivative SLL-0197800, known for its anticoronaviral activities, was tested for its inhibitory effects on RNA-dependent RNA polymerase (RdRp) and 3′-to-5′ exoribonuclease (ExoN) proteins [157]. In vitro assays demonstrated potent inhibitory activities with low EC50 values (0.16 and 0.27 μM, respectively). In silico results supported these findings, indicating the potential of SLL-0197800 against various coronaviral strains, including future versions, such as SARS-emphasising, emphasizing its specific anticoronaviral qualities. This study underscores the potential of drug repurposing and highlights SLL-0197800 as a promising candidate for broader coronaviral infections [157].
In addition, therapeutic options have been sought in natural products (terpenoids, alkaloids, saponins and phenolics) with promising in vitro and in silico results for use in COVID-19 disease [158-160]. Among these, the most studied are resveratrol, quercetin, hesperidin, curcumin, myricetin, and betulinic acid, which were proposed as SARS-CoV-2 inhibitors [159]. Regarding natural products, resveratrol, curcumin, and quercetin have demonstrated in vitro antiviral activity against SARS-CoV-2, and in vivo, a nebulised formulation has been demonstrated to alleviate the respiratory symptoms of COVID-19 [158,159,161]. See line 387 to line 412 in the updated version of the manuscript.
- Paula Andrea Velásquez, Juan C Hernandez, Elkin Galeano, Jaime Hincapié-García, María Teresa Rugeles & Wildeman Zapata-Builes (2024) Effectiveness of Drug Repurposing and Natural Products Against SARS-CoV-2: A Comprehensive Review, Clinical Pharmacology: Advances and Applications, 16:, 1-25, DOI: 10.2147/CPAA.S429064
- Figueredo, J., Lopez, L.F., Leguizamon, B.F., Samudio, M., Pederzani, M., Apelt, F.F., Añazco, P., Caballero, R. and Bianco, H., 2024. Clinical evolution and mortality of critically ill patients with SARS-CoV-2 pneumonia treated with remdesivir in an adult intensive care unit of Paraguay. BMC Infectious Diseases, 24(1), p.37.
Comment 2. Many studies are available on Drug Repurposing with Deep Learning. Please include a literature table highlighting this perspective.
Response to comment 2: Thank you for the constructive comment. The issue has been addressed in the drug discovery section and we have added a paragraph that reads as follows: Artificial intelligence (AI) plays a crucial role in various aspects of addressing the challenges posed by COVID-19, including modelling and simulation, employing AI robotics for medical quarantine, and predicting the spread of the virus. AI systems, encompassing machine learning, deep learning, convolutional neural networks, and cognitive computing, offer significant utility in virus detection, comprehensive screening, continuous monitoring, and alleviating the workload for healthcare providers. Furthermore, they prove instrumental in anticipating potential interactions with novel treatments. In light of these advancements, we recommend that computer scientists focus their efforts on refining and advancing these AI-driven methods, as they hold promise for effective responses to future. See line 584 to line 593 in the updated version of the manuscript.
- Adly AS, Adly AS, Adly MS, Approaches Based on Artificial Intelligence and the Internet of Intelligent Things to Prevent the Spread of COVID-19: Scoping Review J Med Internet Res 2020;22(8):e19104
- Yu K, Beam AL, Kohane IS. Artificial intelligence in healthcare. Nat Biomed Eng 2018 Oct;2(10):719-731.
- McCall B. COVID-19 and artificial intelligence: protecting health-care workers and curbing the spread. Lancet Digit Health 2020 Apr;2(4):e166-e167.
Comment 3. Elaborate in the paper on the major vaccines for SARS-Covid-19, addressing both mutation and normal proteins.
Response to comment 3: Thank you for the constructive comment. The issue has been addressed and reads as follows: Scientists globally engage in a race to develop effective vaccines against SARS-CoV-2, utilizing traditional and next-gen platforms. Current COVID-19 vaccines aim to induce immune responses against the viral spike protein or its subunits, preventing cellular entry. Notably, all FDA-approved vaccines effectively generate neutralizing antibodies. The emergence of SARS-CoV-2 variants poses challenges, prompting the design of a peptide-based vaccine with epitopes targeting pathogenic proteins. Computational methods, reverse vaccinology, and immunoinformatics guide the creation of a multi-epitope-based peptide vaccine, considering mutations in spike glycoprotein. Immune profiling, codon optimization, and in-silico simulations contribute to vaccine development. The Pfizer–BioNTech and Moderna mRNA vaccines, among others, receive global approvals, reflecting ongoing efforts to combat the evolving virus. Several vaccines have been approved in different parts of the world, such as CoronaVac, BBIBP-CorV, CoviVac, Covaxin, Oxford–AstraZeneca vaccine (ChAdOx1 nCoV-19), Sputnik V, the Johnson & Johnson vaccine, Convidicea, RBD-Dimer, and EpiVacCorona.
Subsequently, a study examined the impact of different treatments on SARS-CoV-2 antibody levels in patients with autoimmune rheumatic disease undergoing Pfizer/BioNTech mRNA vaccination [177]. Original adalimumab treatment resulted in stable antibody levels, whereas biosimilar adalimumab showed a significant decrease [177]. This study underscores the dominant influence of treatment type on antibody changes over time, with negligible contributions from other variables [178]. In the context of mRNA vaccines, a booster administered 3–4 weeks after the initial vaccination substantially increased protective antibody levels [179]. Ipsilateral boost of the SARS-CoV-2 mRNA vaccine induced more germinal centre B cells specific to the receptor binding domain (RBD), generating increased bone marrow plasma cells compared with the contralateral boost. Ipsilateral boost also rapidly produced high-affinity RBD-specific antibodies with enhanced cross-reactivity to the Omicron variant [179]. Evaluating T-cell responses in patients with late-stage lung cancer receiving immune-modulating agents, including anti-PD-1/PD-L1, showed distinct CD8+ T-cell responses with only marginal CD4+ T-cell responses to the Omicron variant. This emphasises the need for heightened protective measures for cancer patients because of qualitative deviations in T-cell responses [180]. See lines 413-436 and lines458-473 in the updated version of the manuscript.
Top of Form
Comment 4. Add the structure of the paper at the end of the introduction.
Response to comment 4: Thank you for the constructive comment. The added section reads as follows: This paper contributes to the existing literature on SARS-CoV-2 and is organised as follows: clinical description of SARS-CoV-2, SARS-CoV-2 prevalence and pathology, discussion on postulated hypothesis on COVID-19 mutations, COVID-19 therapies, vaccines, and other ongoing clinical trials therapies, drug repurposing for COVID-19, non-pharmacological interventions, and finally conclusion and future perspectives for COVID-19 research. See lines 90-96 in the updated version of the manuscript
Comment 5. All the images appear to be copied directly. Have you obtained permission to use them in your paper?
Response to comment 5: Thank you for the constructive comment. The images were redrawn and authors cited after modification of original ones using BioRender. The authors allowed the adaptation as long as they are cited as primary sources.
Comment 6. It would be beneficial to include some drugs that are highly predicted for SARS-COVID-19 in recent studies.
Response to comment 6: Thank you for the constructive comment. The mentioned comment has been addressed and reads as follows: The study explores drug repurposing as a cost-effective strategy for SARS-CoV-2 treatment by docking 16 antiviral approved drugs against the protease protein (6M03) responsible for viral replication. Delavirdine, fosamprenavir, imiquimod, stavudine, and zanamivir exhibited excellent results, suggesting their potential as strong reprofile drugs. Isoquinoline derivative SLL-0197800, known for its anticoronaviral activities, was tested for its inhibitory effects on the RNA-dependent RNA polymerase (RdRp) and 3′-to-5′ exoribonuclease (ExoN) proteins. In vitro assays demonstrated potent inhibitory activities with low EC50 values (0.16 and 0.27 μM, respectively). In silico results supported these findings, indicating SLL-0197800's potential against various coronaviral strains, including future versions, such as SARS-CoV-3, emphasizing its nonspecific anticoronaviral qualities. The research underscores the potential of drug repurposing and highlights SLL-0197800 as a promising candidate for broader coronaviral infections. See line 392 to line 404 in the updated version of the manuscript.
Comment 7. Provide a proper introduction to the Covid family, including SARS-CoV, MERS-CoV, SARS-CoV-2, and BAT-CoV, among others.
Response to comment 7: Thank you for the constructive comment. The issue has been addressed as follows: Coronaviruses, a group of RNA viruses, cause various diseases in humans and animals, exhibiting a characteristic crown-like shape under an electron microscope. These viruses are zoonotic, transmitted from animals to humans. Notable among them are SARS-CoV, MERS-CoV, and SARS-CoV-2. SARS-CoV initiated a global outbreak in 2003, with over 8,000 cases and significant mortality. MERS-CoV, identified in 2012, primarily affects the Middle East with a higher mortality rate. SARS-CoV-2, causing COVID-19, emerged in Wuhan, China, in late 2019, leading to a widespread global pandemic. All three viruses share zoonotic origins, posing significant challenges to public health. SARS-CoV and MERS-CoV were managed through measures like quarantine, contact tracing, and travel restrictions, while no specific treatment or vaccine exists for MERS-CoV, making it an ongoing concern, particularly in the Middle East. SARS-CoV-2, the most recent coronavirus, demonstrates a zoonotic origin, possibly from bats and an intermediate host like a pangolin, before infecting humans (20-24, 35 – 40 in Thesis). See line 46 to line 59 in the updated version of the manuscript.
Comment 8. Include analysis studies such as genome analysis and microsatellite analysis in your paper.
Response to comment 8: Thank you for the constructive comment. The issue has been addressed and reads as follows: Genomic and microsatellite analyses identified 1,191 protein targets related to ACE2, revealing 305 host factors associated with COVID-19/asthma comorbidity [73]. Enrichment analyses highlighted the significance of metabolic processes, Th1 and Th2 cell differentiation, and PPAR signalling pathways in this comorbidity. Key host factors, including HRAS, IFNG, CAT, CDH1, FASN, ACLY, CCL5, VCAM1, SCD, and HMGCR, were identified, with some highly expressed in lung tissues [74-76]. In the exploration of genetic variations in the ACE2 and TMPRSS2 receptor genes in COVID-19 patients, there were associations between specific genotypes and expressions of these genes with SARS-CoV-2 positivity, including clinical outcomes [77]. The identification of haplotypes and variants/mutations contributes to the understanding of host genetic factors influencing COVID-19 susceptibility, offering insights for future vaccine development and therapeutic approaches [78,79]. See line 238 to line 249 in the updated version of the manuscript.
Round 2
Reviewer 2 Report
Comments and Suggestions for Authors
Dear Authors,
All my comments are addressed.
Best.
Comments on the Quality of English LanguageMinor editing of English language required